# Estimation equation of limb lean soft tissue mass in Asian athletes using bioelectrical impedance analysis

Yeong-Kang Lai[1], Chu-Ying Ho[1], Ai-Chun Huang[2], Hsueh-Kuan Lu[3], Kuen-Chang Hsieh [4,5]*

1 Department of Electrical Engineering, National Chung Hsing University, Taichung City, Taiwan,
2 Department of Oral Hygiene, Tzu-Hui Institute of Technology, Nanzhou Township, Pingtung County, Taiwan, 3 General Education Center, National Taiwan University of Sport, Taichung City, Taiwan, 4 Big Data Center, National Chung Hsing University, Taichung City, Taiwan, 5 Department of Research & Development, StarBIA Meditek Co., Ltd., Taichung City, Taiwan

* abaqus0927@yahoo.com.tw

## Abstract

**Data Availability Statement:** All relevant data are contained in the paper and its Supporting information files.

### Background

The lean soft tissue mass (LSTM) of the limbs is approximately 63% of total skeletal muscle mass. For athletes, measurement of limb LSTM is the basis for rapid estimation of skeletal muscle mass. This study aimed to establish the estimation equation of LSTM in Asian athletes using bioelectrical impedance analysis (BIA).

### Methods

A total of 198 athletes (121 males, 77 females; mean age 22.04 ± 5.57 years) from different sports in Taiwan were enrolled. A modeling group (MG) of 2/3 (n = 132) of subjects and a validation group (VG) of 1/3 (n = 68) were randomly assigned. Using the InBody S-10, resistance and reactance were measured at 50 kHz from the right palm to the right sole while the participant was in the supine position. Predictor variables were height (h), weight (W), age, Sex, Xc, resistance index (RI; RI = $h^2$ / R). LSTM of arms and legs measured by dual-energy X-ray absorptiometry (DXA) was the response variable. Multivariate stepwise regression analysis method was used to establish BIA estimation equations as $ArmsLSTM_{BIA-Asian}$ and $LegsLSTM_{BIA-Asian}$. Estimation equations performance was confirmed by cross-validation.

### Results

Estimation equation "$ArmsLSTM_{BIA-Asian}$ = 0.096 $h^2$/R– 1.132 Sex + 0.030 Weight + 0.022 Xc– 0.022 h + 0.905, $r^2$ = 0.855, SEE = 0.757 kg, n = 132" and "$LegsLSTM_{BIA\ Asian}$ = 0.197$h^2$/R" + 0.120 h– 1.242 Sex + 0.055 Weight– 0.052 Age + 0.033 Xc –16.136, $r^2$ = 0.916, SEE = 1.431 kg, n = 132" were obtained from MG. Using DXA measurement results of VG for correlation analysis and Limit of Agreement (LOA) of Bland-Altman Plot, ArmsLST is 0.924, -1.53 to 1.43 kg, and LegsLST is 0.957, -2.68 to 2.90 kg.

**Funding:** This work was supported by grants from the Tzu Hui Institute of Technology Research Program (THIT-110004) and was supported in part by the National Science and Technology Council, Republic of China, under Grants MOST 111-2622-E-005-001. The funders had no role in the design of the study; in the collection, analysis, or interpretation of the data; in writing the manuscript or in the decision to publish the results.

## Conclusion

The established single-frequency BIA hand-to-foot (whole body) estimation equation quickly and accurately estimates LSTM of the arms and legs of Asian athletes.

## Introduction

In the field of sports science, research on the body composition of athletes has developed vigorously in recent years. Many studies have classified athletes by different phenotypes, including shape, weight, body fat percentage, fat mass, fat-free mass (FFM), lean soft tissue mass (LSTM), muscle mass (MM) and other body components. Due to different sports categories, sex, and competitive levels, early research focused on body fat percentage [1]. Current related research has been extended to total and regional skeletal muscle mass in athletes. More recently, these measurements, especially LSTM, are related to improving athletic performance and reducing the risk of injury [2]. However, assessments of body composition in individual athletes still need to be targeted as factors for problem-solving among athletes. In this way, individual athletes can obtain corresponding personal benefits when measuring body composition [3]. This innovative approach goes beyond previous fundamental issues of body composition and emphasizes movement-specific performance. For example, the characteristics of the body composition of arms, legs and torso are directly related to training progress [4].

The latest development of body composition technology divides the body into three-components, namely LSTM, FM and bone mineral content (BMC). LSTM includes whole body water, protein, carbohydrates, non-fat lipids, and soft tissue minerals [5]. The three-component body composition model can be used with dual-energy X-ray absorptiometry (DXA), magnetic resonance imaging (MRI), and computerized tomography (CT). Although CT and MRI provide information on muscle cross-sectional areas (CSA) and muscle volume (MV), these methods are costly, require relatively in-depth research expertise, and analysis of the results can be difficult. Therefore, these methods are not practical in the sports performance setting. Compared with the above methods, although the cost of DXA is lower, it still has limitations in application fields. Another measurement method used commonly in the two component model is bioelectrical impedance analysis (BIA). Because BIA is convenient, safe, non-invasive, and fast, it provides athletes with instant information on body composition measurements, and LSTM changes can be tracked conveniently during athlete training.

In women's softball [6], men's hockey [7] and soccer [8], the variability of the whole body and limb segment was the least. Among male college basketball players, centers players reflect that their arms and legs have the highest LSTM and FM compared to players in all other positions (with the exception of power forwards) [9]. American football players also have the same body composition as men's basketball, with significant differences between the different positions [10, 11]. Results of these studies suggest that there may be a relationship between limb segments, body composition and exercise programs, positions, and specific functional needs (e.g., shooting, competitive season of hockey (and likely in football and other sports), the LSTM of the legs of excellent players will increase significantly as the season time increases [12]. For football players, the LSTMs on the legs and torso are also noted to increase significantly as the season progresses [13]. These studies have shown that body composition changes over time.

Since 1990, several important studies have been published on the use of BIA for estimates body composition in athletes [14, 15]. The measurement, comparison and validation studies of commercial BIA devices in LSTM or FFM for the whole body and limbs of athletes have been

proposed and have gradually captured increasing attention [16–18]. The physical characteristics of athletes are different from those of the general population based on long-term trends in specific sports [19]. Therefore, special mechanisms must be applied or dedicated regression equations to accurately measure body composition [20]. Recently, body composition estimation equations have been proposed to measure the body composition of athletes using BIA to distinguish Sex differences [21, 22]. A recent study by Sardinha *et al.* [23] proposed a BIA estimation equation for athletes using LSTM for upper and lower limbs, although the subjects of this study were Europeans and Americans. Published research has already shown ethnicity differences in the estimation of body composition by BIA [24]. In the past, there were some research on the measurement equation of FFM in athletes [14, 22, 25, 26], and recently there was research on the establishment of equations of LSTM in Caucasian athletes [23]. For athletes in the Asian region, the relevant research was very limited [27]. More cross-ethnic population studies on body composition are warranted. These studies should be meticulously conducted, employing suitable methodologies and standardization, to ensure the value of the gathered information [28]. Whether applicable or not, it is still necessary to further explore or establish a suitable LSTM estimation equation. Therefore, this study aimed to establish and verify the estimation equation of limb lean soft tissue mass by using BIA in Asian athletes. A single-frequency BIA hand-to-foot (whole body) estimation equation was established to quickly and accurately estimates the lean soft tissue mass of the arms and legs of Asian athletes.

## Material and methods

### Study design and sample

The subjects of this prospective observational study were all active high-level athletes in Taiwan and recruited between June 1, 2021 and March 30, 2022. Eligible participants engaged in at least 12 hours of physical or specialty training activities per week. Professional training time was 9.6 ± 2.5 years and received high-intensity or professional training for 12.3 ± 4.5 hours per week. They were all active players at the highest level in Taiwan and in the Asian or Olympic Games (except for dance specialties). Subjects did not drink alcoholic beverages 48 hours before the test, did not use diuretics for 7 days before the test, and did not participate in intense training 24 hours before the test. Females with menstrual periods were excluded [29]. All subjects had no history of nutritional, endocrine or growth disorders and had no more than 5 kg body weight change before the six experiments (Fig 1).

### Ethical considerations

This study was conducted in the Department of Radiology, Dali Jen-Ai Hospital, Taichung City. Before the six experiments, the research protocol and experimental procedure were approved by the Human Experiment Committee of Taso-Tun Psychiatric Center, Ministry of Health and Welfare (IRB-11001). Before the tests began, the research assistant explained the experimental precautions to the subjects. Once the subjects understood the experimental process and their rights, they agreed to participate in the experiment and each subject provided signed informed consent. Informed consent was obtained from each participant and/or guardian if under the age of legal consent prior to testing. We conducted experiments in accordance with relevant guidelines and regulations.

### Procedure and anthropometry

Three days before the experiment, the subjects were informed of the precautions. The subjects reported to the experimental site at 1:00 p.m. on the day of the experiment, and there is no

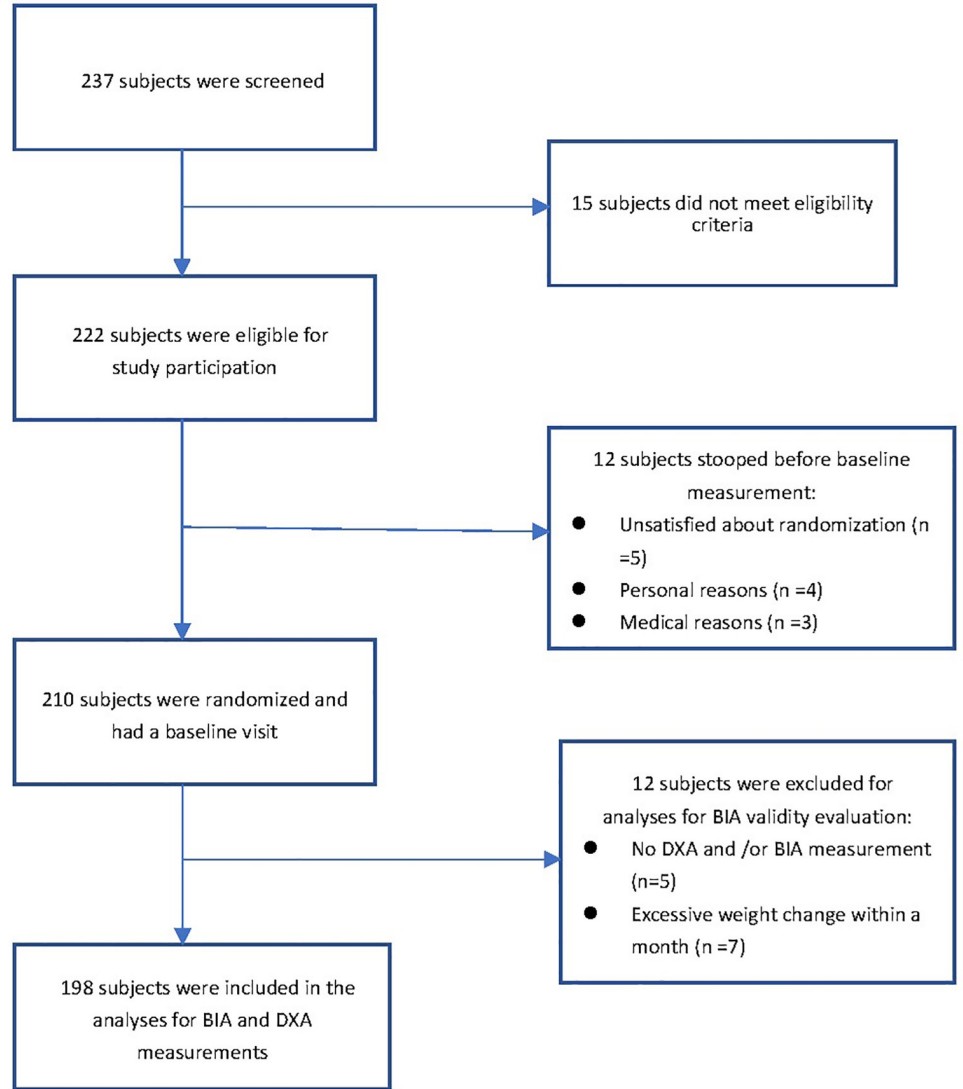

**Fig 1. Flow chart of participants for the inclusion in the analysis for the study.**

lunch to maintain a fasting state before measurement. Filled out the questionnaire on basic personal information and training process, and changed into cotton robes for testing. The bladder of each subject was emptied 20 minutes before the experiment. Subjects were measured using Tanita BC-418MA (Tanita Co., Tokyo, Japan) with an accuracy of 0.1 kg. A height ruler (Holtain, Crosswell, Wales, UK) was used to measure the subjects' height without shoes, providing accuracy of 0.5 cm. Body mass index (BMI) was calculated by dividing the subjects' individual body weight by the square of the height in kg/m$^2$. A cloth tape measure was used to measure subjects' waist and hip circumferences. When measuring waist circumference, subjects placed feet together, relaxed the abdominal muscles, placed arms naturally at each side of the body, breathed normally, and allowed measurement of the narrowest position of the body below the ribs and above the navel. Hips were measured at the widest point. Waist and hip circumferences were measured twice, and the average values were recorded.

## Impedance measurement

Resistance and reactance were measured from the dorsum of the right palm to the dorsum of the right foot using a bioimpedance analyzer, the S-10 (InBody Co., Ltd., Seoul, South Korea), operating at a frequency of 50 kHz while the participant was in a supine position. The positioning of the electrodes and body posture followed the instructions provided by the electrode manufacturer (BIATRODES® Akern Srl; Florence, Italy). The InBody S-10 is a Class II medical device approved by the FDA (Regulation Number: 870.2770). Resistors supplied by the manufacturer were used for calibration every day before the experiment. Bioimpedance measurement is done after confirming that accuracy of the bioimpedance measuring instrument meets the requirements. The subject lies on his/her back on a hospital bed. The right wrist and the back of the right foot were connected with silver electrode patches, which were the pair of receiving and transmitting electrodes, respectively. Subjects lay on their backs calmly for five minutes before the measurement was taken. The resistance index (RI) was defined as the ratio of the square of the height to the measured resistance at a frequency of 50 KHz ($h^2/R$). For 5 male and 5 female subjects, impedance measurements were repeated 10 times within one hour of the day. Impedance measurements were performed by 5 subjects for each of the same male and female subjects in the same period for 4 consecutive days. The coefficient of variation (CV = standard deviation/mean × 100%) of impedance, resistance, reactance, and phase angle measurements was evaluated for within-day and between-day results, respectively. CVs were 0.2%-0.7%, 0.1%-0.8, 0.3–1.2%, 0.3–1.4% and 0.8%-1.6%, 0.9%-2.1%, 0.8%-1.7%, 1.0%-2.2% respectively.

## Dual energy X-ray absorptiometry

Body composition was measured by DXA (Lunar Prodigy; GE Medical Systems, Madison, WI, USA) and the Microsoft software, Encore 2003 Version 7.0. The subjects performed the DXA measurement within thirty minutes immediately after the impedance measurement was completed. Each subject wore a light cotton robe and lay on the measuring bed in a relaxed supine position. The upper limbs were stretched and placed on both sides of the body, with toes facing upward and feet slightly side by side. The procedure took about 20 minutes for each test subject. Complete DXA measurement of body composition included LSTM, FM (fat mass), BMC (bone mineral content) on the whole body, left and right upper limbs, left and right lower limbs, trunk and other parts. The regions of interest (ROI) for the whole body, trunk, legs, arms, android, and gynoid were automatically determined by the software.

## Statistical analysis

All data were expressed as means ± SDs, and range. Data were tested for normal distribution using the Kolmogorov-Smirnov test. All statistical analyses were performed using SPSS Ver.20 (Statistical Package for the Social Sciences, IBM SPSS statistics for Windows; IBM Corp., Armonk, NY, USA). The significance level was set at $p < 0.05$ (two-tailed). Given that a sample size of 124 participants was calculated considering a type 1 error of 5% and a power of 80% to achieve a moderate effect size for the $R^2$ increases in the prediction equation with the inclusion of five predictors, our sample size of 198 athletes was sufficient for assuring an adequate power analysis in model development.

Paired-$t$ tests were used to compare DXA and BIA on ArmsLSTM and LegsLSTM. The correlation between BIA estimates and DXA-measured ArmsLSTM and LegsLSTM was performed using Pearson's correction and Lin's concordance correlation coefficient (CCC). Bland-Altman plots were used to represent the mean difference and limit of agreement (LOA). The regression line was used to represent the trend.

Two-thirds ($n$ = 132) and one-third ($n$ = 66) of the total subjects were randomly divided into a modeling group (MG) and a validation group (VG), respectively. Height, weight, age, Sex, resistance index, and reactance were used as predictor variables, and the LSTM of upper limbs (arms) and lower limbs (legs) measured by DXA were used as response variables, which were expressed as ArmsLSTM$_{DXA}$ and LegsLSTM$_{DXA}$, respectively. Stepwise regression analysis was performed with MG. Set parameters—forward (F$_{in}$ = 4.00), backward (F$_{out}$ = 3.99) to obtain the selected predictor variable. The resulting equations were ArmsLSTM$_{BIA-Asian}$, LegsLSTM$_{BIA-Asian}$ and their corresponding regression coefficients, standard estimate error (SEE), coefficient of determination ($r^2$), and predictor variables removed with VIF (variance inflation factor) > 4. The ArmsLSTM$_{BIA-Asian}$ and LegsLSTM$_{BIA-Asian}$ obtained by the VG data were brought into the MG, and were analyzed by correlation and Bland-Altman Plots, respectively, to evaluate the effectiveness of the LSTM estimation equation. Intraclass correlation coefficient (ICC) analysis was conducted using two repeated measurements to assess the reproducibility of estimating ArmLSTM and LegLSTM in this study. The measurement unit was kilograms (kg).

## Results

A total of 198 male and female athletes participated in this study, including 121 male athletes (basketball: 11, swimming: 18, powerlifting: 8, judo: 10, long-distance running: 4, football: 26, wrestling: 37, track and field: 7), and 77 female athletes (basketball: 7, dance: 10, judo: 15, long-distance running: 2, tug-of-war: 9, soccer: 17, wrestling: 2, track and field: 15). The age, height, weight and body fat percentage of male athletes were 22.67 ± 5.82 years, 174.87 ± 8.25 cm, 74.9 ± 11.87 kg and 15.26 ± 6.83%, respectively. Fig 1 shows the flow chart of participants included in the study analysis. For female athletes, mean age, height, weight, and body fat percentage were 21.02 ± 5.00 years, 161.64 ± 6.86 cm, 58.15 ± 9.35 kg and 27.15 ± 8.06%, respectively. For all subjects, MG and VG personal characteristics parameters and body composition data are shown in Table 1.

In the present study, all subjects' data were entered into the arms LSTM equation of Sardinha *et al.* [23] and compared with the present DXA measurement results. The scatter plots and regression line are shown in Fig 2a. The LOA and trend line of the two Bland-Altman Plots are shown in Fig 2b. The same procedure and data were entered into the lower limb LSTM equation and compared with the present DXA measurement results. The distribution plots and regression line are shown in Fig 2c. The LOA and trend line of the two Bland-Altman Plots are shown in Fig 2d. Fig 2a and 2c are scatter plots of ArmLSTM and LegLSTM estimates compared to DXA results using the Sardinha equation. The regression line (thin solid line) shows a weak fit with the equivalence line (dashed line), indicating that the accuracy of the Sardinha equation for estimating ArmLSTM and LegLSTM in Asian athletes is limited. The Bland-Altman plots in Fig 2b and 2d show the limits of agreement (LOA) for ArmLSTM and LegLSTM estimates using the Sardinha equation in Asian athletes. The trend line (dashed line) in the plots indicates a significant proportional bias between the two measurement methods.

The estimated variables selected in the stepwise regression analysis method, along with the corresponding regression coefficients, intercept, SEE, determination coefficient, variance inflation factor, and standardized coefficient of the estimation equation, are shown in Tables 2 and 3. In the stepwise regression analysis method, ArmLSTM was used as the response variable. Sequentially selected variables entered into the measurement equation using multivariate regression analysis were resistance index ($h^2$/R), Sex (female = 0, male = 1), W (weight), Xc (resistance), and h (height). The increase in predictor variables and the corresponding

**Table 1. Physical characteristics of the subjects [1].**

| All Subject | Male (*n* = 121) | Female (*n* = 77) | Total (*n* = 198) |
|---|---|---|---|
| Age(year) | 22.7±5.8(17.0, 46.0) | 21.0±5.0(17.0, 45.0) | 22.0±5.6(17.0, 46.2) |
| Height(cm) | 174.9±8.3(155.9, 198.3) | 161.6±6.9(147.1, 181.9)[3] | 169.7±10.1 (147.1, 198.3) |
| Weight(kg) | 74.9±11.9(48.9, 123.9) | 58.2±9.4(40.8,84.7)[3] | 68.4±13.7 (40.8, 123.9) |
| BMI(kg/m$^2$) | 24.5±3.0(18.9, 36.0) | 22.2±2.5 (17.7, 29.8)[2] | 23.6±3.1(17.7, 36.0) |
| R(ohm) | 483.4±52.2(339.8, 696.9) | 602.5±54.7(452.0, 727.0)[3] | 529.7±83.1(339.8, 724.0) |
| Xc(ohm) | 64.7±8.9(43.4, 92.9) | 70.9±7.4(55.8, 86.7)[3] | 67.2±8.9(43.4, 92.9) |
| ArmsLSTM(kg) | 7.0±1.4(2.7, 11.1) | 3.8±0.8 (2.6,7.5)[3] | 5.7±2.0(2.6, 11.1) |
| LegsLSTM(kg) | 22.6±3.4(12.9, 33.7) | 15.0±2.7(10.9,27.5)[3] | 19.6±4.9(10.9,33.7) |
| TrunkLSTM(kg) | 26.4±3.6(14.5–41.7) | 17.8±2.1(13.8,27.1)[3] | 22.8±5.3(13.8, 41.7) |
| FM% | 15.3±6.8(5.10–34.0) | 27.2±8.1(9.40,46.5)[3] | 20.2±9.4(5.1, 46.5) |
| WHR | 0.82±0.05(0.72,0.96) | 0.80±0.06(0.65,0.94)[3] | 0.81±0.05(0.65,0.96). |
| **Modeling Group** | **Male (*n* = 70)** | **Female (*n* = 52)** | **Total (*n* = 132)** |
| Age(year) | 22.9±6.1(17.0, 46.0) | 20.9±5.0(17.0, 42.0)[2] | 22.1±5.8(17.0, 46.0) |
| Height(cm) | 174.8±7.8(157.3, 198.3) | 160.9±6.4(150.3, 181.9)[3] | 169.3±10.0(150.5, 198.3) |
| Weight(kg) | 73.7±9.9(56.9,123.1) | 57.4±8.7(40.8, 84.7)[3] | 67.3±12.4(40.8, 123.1) |
| BMI(kg/m$^2$) | 24.1±2.7(18.9, 36.0) | 22.1±2.5(17.7, 29.8)[2] | 23.3±2.9(17.7, 36.0) |
| R(ohm) | 483.8±58.8(339.8,679.5) | 602.3±49.8(506.2, 704.7)[3] | 530.5±80.2(339.8, 704.7) |
| Xc(ohm) | 65.2±8.8(43.4, 89.9) | 70.7±7.5(55.8, 86.8)[3] | 67.3±8.7(43.4, 89.9) |
| ArmsLSTM(kg) | 6.9±1.3(2.6, 9.4) | 3.8±0.7(2.6, 6,8)[3] | 5.7±1.9(2.6, 9.4) |
| LegsLSTM(kg) | 22.5±3.0(15.4, 32.6) | 14.8±2.4(10.9, 25.7)[3] | 19.4±4.7(10.9, 32.6) |
| TrunkLSTM(kg) | 26.3±3.6(14.5,41.7) | 17.7±2.1(13.9,27.1)[3] | 22.7±5.2(13.9,41.7) |
| BF% | 15.2±7.1(5.4,33.7) | 27.5±8.5(9.4,46.5)[3] | 20.4±9.8(5.4,46.5) |
| WHR | 0.81±0.05(0.72,0.96) | 0.79±0.06(0.69,0.92)[3] | 0.81±0.05(0.69,0.96) |
| **Validation Group** | **Male (*n* = 41)** | **Female (*n* = 25)** | **Total (*n* = 66)** |
| Age(year) | 22.1±5.3(18.0, 46.0) | 21.3±5.2(17.0, 45.0)[2] | 21.8±5.2(17.0, 46.0) |
| Height(cm) | 175.0±9.1 (155.8, 194.6) | 163.3±7.6(147.1, 178.1)[3] | 170.6±10.3(147.1, 194.6) |
| Weight(kg) | 77.4±18.8 (48.9, 119.1) | 59.7±10.6(45.2, 81.2)[3] | 70.7±15.9(45.2, 119.1) |
| BMI(kg/m$^2$) | 25.2±3.6(19.6, 33.7) | 22.3±2.6(18.5, 29.4)[2] | 24.1±3.5(18.5, 33.7) |
| R(ohm) | 482.6±69.2(355.2, 696.9) | 602.8±64.8(452.0, 724.0)[3] | 528.1±89.2(355.2, 724.0) |
| Xc(ohm) | 63.8±9.3(43.4, 92.8) | 71.4±7.1(56.8, 83.7)[3] | 66.7±9.2(43.4, 92.8) |
| ArmsLSTM(kg) | 7.1±1.6(3.0, 11.1) | 3.9±1.1(2.6, 7.5)[3] | 5.9±2.1(2.6, 11.1) |
| LegsLSTM(kg) | 22.7±4.2 (12.9, 33.7) | 15.5±3.4(11.9, 27.5)[3] | 20.0±5.3(11.9, 33.7) |
| TrunkLSTM(kg) | 26.6±3.8(19.7,36.8) | 17.8±1.9(13.8,23.0)[3] | 22.9±5.4(13.8,36.8) |
| FM% | 15.3±6.4(5.10,34.0) | 26.4±7.1(13.5,40.7)[3] | 20.0±8.6(5.1, 40.7) |
| WHR | 0.82±0.04(0.74,0.84) | 0.80±0.06(0.65,0.94)[3] | 0.81±0.05(0.65,0.94) |

[1] All values are x ± SD; minimum and maximum in parentheses.

[2,3] Significantly different from male (one-factor ANOVA): [2] $P = 0.005$, [3] $P < 0.001$; BMI, body mass index; R, resistance; Xc, reactance, LSTM, Lean soft tissue mass, WHR, Waist-to-hip ratio; FM%, Fat Mass % (body fat percentage).

coefficient of determination, standard error of the estimate (SEE), standardized coefficient (β), and variance inflation factor (VIF) are shown in Table 2. The best estimation equation for ArmsLSTM is as follows:

$$\text{ArmsLSTMB}_{\text{IA}-\text{Asian}} = 0.096\text{h}^2/\text{R} - 1.132\,\text{Sex} + 0.030\,\text{W} + 0.022\,\text{Xc}$$
$$-0.022\,\text{h} + 0.905, (\text{r}^2 = 0.855, \text{SEE} = 0.757\,\text{kg}, \text{n} = 132, \text{p} < 0.01) \tag{1}$$

where: h$^2$/R, resistance index; W, weight; Xc, reactance; h, height;

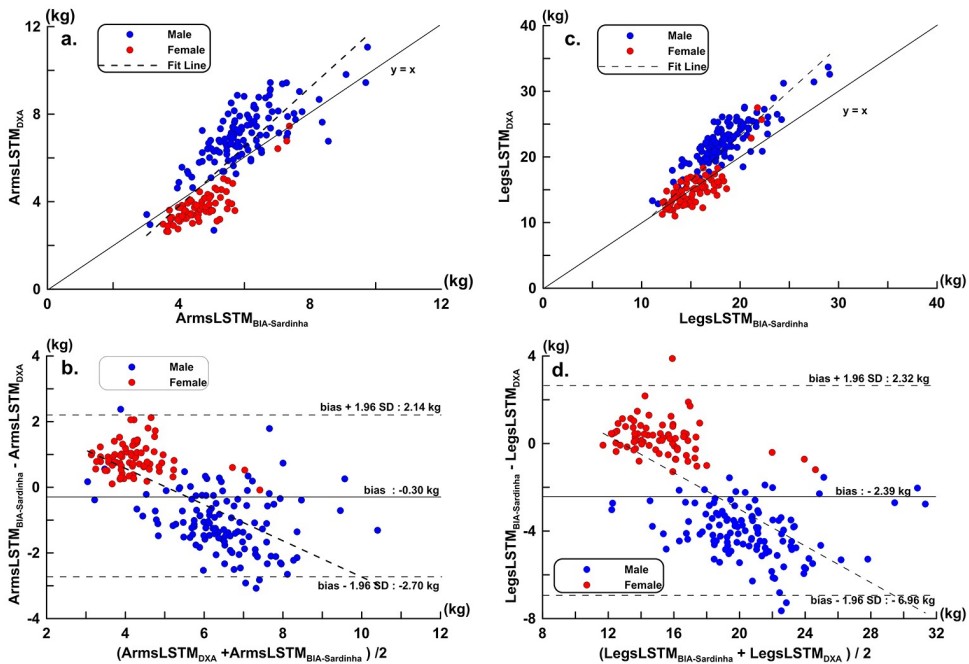

**Fig 2. Asian athletes were compared with DXA using the Sardinha equation.** (**a**) Scatter plots and regression line of ArmsLSTM (y = 1.358 x– 1.650, r = 0.812, p < 0.001, CCC = 0.784); (**b**) Bland-Altman Plots of ArmsLSTM (trend, y = −0.551 x + 2.780, p < 0.001); (**c**) Distribution plot and regression line of LegsLSTM (y = 1.359 x -3.941, r = 0.863, p < 0.001, CCC = 0.824); (**d**) Bland-Altman Plots of LegsLSTM (trend, y = −0.418 x + 6.714, p < 0.001).

Using the same measurement variables as for MG, LegsLSTM was the response variable, and applying multivariate stepwise regression analysis, the sequentially selected variables entered the measurement equation were $h^2$/R, h, Sex, W, Age, and Xc. The corresponding coefficients of determination, SEE, VIF, and β are shown in Table 3. The optimal estimation equation of LegsLSTM is as follows:

$$\text{LegsLSTM}_{\text{BIA}-\text{Asian}} = 0.197 h^2/R + 0.120\,h - 1.242\,\text{Sex} + 0.055\,W - 0.052\,\text{Age} + 0.033\,\text{Xc} - 16.136, (r^2 = 0.916, \text{SEE} = 1.431\,\text{kg}, n = 132; p < 0.001) \tag{2}$$

**Table 2. Multiple regression analysis results for ArmsLSTM, based on bioimpedance index and anthropometric.**

| ArmsLSTM, Modeling Group (n = 132) | | | | | | | | | |
|---|---|---|---|---|---|---|---|---|---|
| $h^2$/R | + Sex | +Weight | +Xc | +h | Intercept | SEE | $r^2$ | VIF | β |
| -0.131 ± 0.005** | - | - | - | - | -0.663 ± 0.270 ** | 0.878 | 0.802 | 2.27 | 0.659 |
| 0.100 ± 0.006** | -1.147 ± 0.171** | - | - | - | -0.547 ± 0.418 | 0.797 | 0.837 | 2.39 | -0.281 |
| 0.094 ± 0.426** | -1.184 ± 0.171** | 0.026 ± 0.007** | - | - | 0.094 ± 0.426* | 0.775 | 0.847 | 3.57 | 0.211 |
| 0.087 ± 0.010** | -1.102 ± 0.172** | 0.025 ± 0.007** | 0.019 ± 0.008** | - | -1.742 ± 0.871 | 0.765 | 0.848 | 1.66 | 0.098 |
| 0.096 ± 0011** | -1.132 ± 0.171** | 0.030 ± 0.007** | 0.022 ± 0.008** | -0.022 ± 0.010** | 0.905 ± 1.462* | 0.757 | 0.855 | 3.29 | -0.112 |

ArmsLSTM, Arms lean soft tissue mass; Regression coefficient estimate ± SEE; $r^2$, coefficient of determination;

* p < 0.01;

** p < 0.001;

β, standardized coefficient; VIF: variance inflation factor; SEE, standard error of the estimate; h, height.

**Table 3. Multiple regression analysis results for LegsLSTM, based on bioimpedance index and anthropometric.**

| | | | | | | LegsLSTM, Modeling Group (n = 132) | | | | |
|---|---|---|---|---|---|---|---|---|---|---|
| $h^2$/R | +h | + Sex | +Weight | +Age | +Xc | Intercept | SEE | $r^2$ | VIF | β |
| .336±.010** | - | - | - | - | | .675±.554 | 1.792 | .863 | 2.89 | .546 |
| .241±.014** | .156±.019** | - | - | - | | -20.496±2.577** | 1.543 | .899 | 3.35 | .248 |
| .213±.016** | .150±.018** | -1.151±.332** | - | - | | -17.497±2.653** | 1.502 | .905 | 2.42 | -.125 |
| .179±.018** | .132±.019** | -1.254±.325** | .050±.014** | - | | -15.845±2.263** | 1.467 | .911 | 3.60 | .155 |
| .178±.018** | .127±.018** | -1.362±.323** | .053±.014** | -.048±.019** | | -14.030±2.683 | 1.443 | .914 | 1.04 | -.059 |
| .197±.020** | .120±.019** | -1.242±.324** | .055±.014** | -.052±.019** | .033±.015* | -16.139±2.826 | 1.431 | .916 | 1.67 | .060 |

LegsLSTM,Legs lean soft tissue mass; Regression coefficient estimate ± SEE; $r^2$, coefficient of determination;

* $p < 0.01$;

** $p < 0.001$;

β, standardized coefficient; VIF: variance inflation factor; SEE, standard error of the estimate; h, height

where: $h^2$/R, resistance index; W, weight; Xc, reactance; h, height; Age, age; h, height; Sex (female = 0, male = 1);

Tables 2 and 3 have shown the choosing estimation variables and their orders when evaluating ArmsLSTM and LegsLSTM. VG data are entered into the formula (1) to obtain ArmsLSTM$_{BIA-Asian}$, and compared with the ArmsLSTM$_{DXA}$. The scatter diagram of ArmsLSTM$_{BIA-Asian}$, ArmsLSTM$_{DXA}$, regression line, Bland-Altman Plots and LOA calculation, were drawn, respectively, as Fig 3a and 3b. VG data entered into equation (2) to obtain LegsLSTM$_{BIA-Asian}$, and its scatter and Bland-Altman Plots are shown in Fig 3c and 3d, respectively. Fig 3a and 3c show scatter plots of ArmLSTM and LegLSTM estimates compared to DXA results for the validation group. The regression line (thin solid line) closely aligns with the equivalence line (dashed line). The Bland-Altman plots in Fig 3b and 3d represent the ArmLSTM and LegLSTM estimation equations established by the model group. This indicates no proportional error in the validation group's estimated results. The limits of agreement (LOA) in Fig 3b and 3d are significantly narrower than those in Fig 2b and 2d.

For males, females, and overall subjects, the resistance, reactance, and anthropometric parameters corresponded to the equations by Sardinha *et al.* [23]. The data of ArmsLSTM$_{BIA-Asian}$, LegsLSTM$_{BIA-Asian}$ and DXA in this study are shown in Table 4.

Shown in the Table 5, the correlation between ArmsLSTM and estimated variables was $h^2$/ R ($r = 0.895$), Weight ($r = 0.794$), Height ($r = 0.725$), R ($r = -0.836$), Sex ($r = -0.795$) according to the level of correlation. LegsLSTM was $h^2$/ R ($r = 0.929$), height ($r = 0.864$), Weight ($r = 0.839$), R ($r = -0.777$), and Sex ($r = -0.757$). The estimates of error of ArmsLSTM and LegsLSTM were 0.171 and 0.582 kg. Reproducibility was 0.98, 0.99 respectively.

## Discussion

The present study established BIA estimation equations of LSTM for arms and legs for Asian athletes in Taiwan. Accordingly, this study is the first to compare the differences and applicability of BIA to LSTM estimation equations for Asian and European athletes. Cross-validation results showed that the BIA estimation equation established for athletes' LSTM in this study has good performance and can be applied practically to the limb LSTM measurement of Asian athletes.

Through the measurement of limb LSTM, measuring whole-body skeletal muscle mass can be estimated indirectly, overcoming the difficulty associated with other whole-body

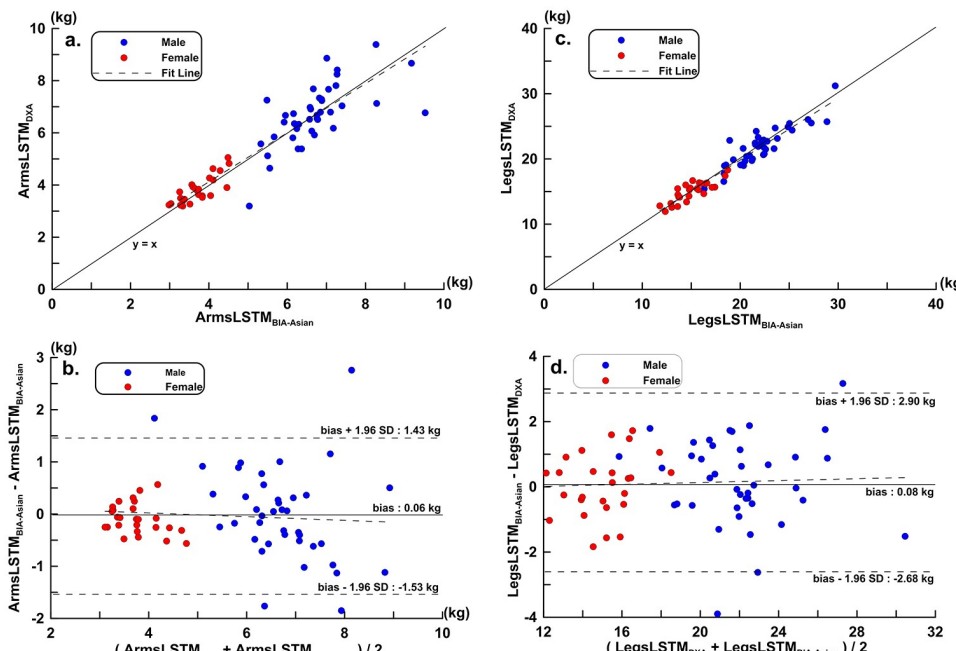

**Fig 3. A new equation was developed to measure the outcomes of Asian athletes compared with DXA in the validation group (VG).** a. scatter plots and regression line of ArmsLSTM (y = 0.024 + 1.007 x, $r$ = 0.924, p < 0.001, CCC = 0.913); b. Bland-Altman Plots of ArmsLSTM (Trend: y = 0.441–0.088 x, p > 0.05); c. Scatter plots and regression line of LegsLSTM (y = -0.094 + 1.000 x, $r$ = 0.957, p < 0.001, CCC = 0.943); d. Bland-Altman Plots of LegsLSTM (trend: y = 0.978–0.04 x, p > 0.05).

measurement methods. Accurately measuring the LSTM of the limbs is exceptionally important for measuring the whole-body skeletal muscle mass of athletes [30]. This study used data measured from Asian athletes and applied the equation of Sardinha et al. For the upper limbs, the LSTM of female athletes was generally overestimated, while the LSTM of male athletes was slightly underestimated. For the lower limbs, the LSTM of male athletes was significantly underestimated. These results showed clear proportional effects, with greater underestimation as the LSTM of the upper and lower limbs increased. Therefore, the applicability of the LSTM

**Table 4. LegsLSTM and ArmsLSTM estimation equation by DXA and bioimpedance for Asian Athlete[1].**

| Method | Male (*n* = 121) | Female (*n* = 77) | Total (*n* = 198) |
|---|---|---|---|
| | | ArmsLSTM | |
| DXA(kg) | 6.99 ± 1.38 (2.69, 11.06) | 3.79 ± 0.82 (2.6,7.5) ** | 5.74 ± 1.97 (2.6, 11.1) |
| Sardinha [14] (kg) | 5.93 ± 1.13 (3.03,9.75) | 4.67 ± 0.76 (3.52, 7.37)** | 5.44 ± 1.18 (3.14,7.,79) |
| Asian(kg) | 6.92 ± 1.06 (4.25, 10.30) | 3.74 ± 0.65 (2.72, 6.14) | 5.74 ± 1.81 (2.14,9.92) |
| | | LegsLSTM | |
| DXA (kg) | 22.55 ± 3.42 (12.86, 33.70) | 15.03 ± 2.74 (10.9,27.5) ** | 19.62 ± 4.85 (10.9,33.7) |
| Sardinha [14] (kg) | 18.58 ± 2.91 (11.09,29.11) | 15.11 ± 2.30(11.24,22.97)** | 17.33 ± 3.06 (11.34,23.45) |
| Asian (kg) | 22.62 ± 3.14 (14.12,31.72) | 15.03 ± 2.74 (10.98, 27.49) ** | 19.62 ± 4.64 (10.61,28.98) |

[1] All values are x ± SD; minimum and maximum in parentheses. Significantly different from male (one-factor ANOVA):

*, $P$ = .05,

**, $P$ < .01

**Table 5. Correlation coefficient matrix of independent and dependent variables (_n_ = 198).**

|  | ArmsLSTM | LegsLSTM | Sex | Wegith | Xc | R | h²/R | Age | Height |
|---|---|---|---|---|---|---|---|---|---|
| **ArmsLSTM** | 1 | .908** | -.795** | .794** | -.451** | -.836** | .895** | .122 | .725** |
| **LegsLSTM** |  | 1 | -.757** | .839** | -.487** | -.777** | .929** | .034 | .864** |
| **Gender** |  |  | 1 | -.601** | .341** | .701** | -.748** | -.145* | -.642** |
| **Weight** |  |  |  | 1 | -.500** | -.687** | .833** | .106 | .765** |
| **Xc** |  |  |  |  | 1 | .624** | -.597** | .017 | -.399** |
| **R** |  |  |  |  |  | 1 | -.900** | -.133 | -.517** |
| **h²/R** |  |  |  |  |  |  | 1 | .093 | .811** |
| **Age** |  |  |  |  |  |  |  | 1 | .028 |
| **Height** |  |  |  |  |  |  |  |  | 1 |

where: $h^2/R$, resistance index; Xc, reactance; R, resistance; h, height;

* $p < 0.05$;

** $p < 0.001$

estimation equation established for European athletes to Asian athletes must be carefully evaluated. Therefore, some scholars have conducted validation studies on the LSTM or FFM measurements of athletes' limbs for commercial BIA devices. Brewer _et al._ [16] used DXA as a standard for comparison among male athletes at Division I College in the United States. The results showed that the LOA of Inbody770 measured by FFM (free fat mas, FFM) of the arms and legs of male athletes was −3.1 to 0.5 kg, −15.3 kg to 2.2 kg, respectively. Female athletes were −1.5 to 0.5 kg, −5.9 to 0.4 kg [18]. Raymond _et al._ [17] also compared the DXA of male college football players, and the correlation coefficients between the LOA and the FFM of the upper and lower extremities of the men's athletes in Inbody770 were −0.43 to 3.23 kg ($r = 0.82$), 0.73 to 9.97 kg ($r = 0.78$) [17]. The results of Esco _et al._ [18] showed that the LOA of Inbody720 in upper and lower extremity FFM of female college athletes was −0.74 to 0.84 kg ($r = 0.89$) and −3.03 to 2.21 kg ($r = 0.83$), respectively [18]. Most existing commercial multi-limb bioimpedance analyzers are designed for use in a standing position and are equipped with stainless steel electrode plates to measure the resistance and reactance of the limbs and trunk at different frequencies. In this study, the resistance and reactance of a single section from the dorsum of the right palm to the dorsum of the right foot were measured in a supine position using electrodes. The resistance and reactance measurement environments required for the Sardinha equation compared in this study were also supine with adhesive electrodes. Although different from the BIA device (BIA 101 Anniversary AKERN/RJL Systems; Florence, Italy) used by Sardinha et al. [23], the InBody S-10 used in this study underwent standard resistance and reactance correction procedures. This ensures the accuracy of the measured resistance and reactance. Almost all of the above studies have expressed that the current commercial BIA multi-limb body composition analyzer has errors and limited correlation in the measurement results of bone mineral content of upper and lower extremities and FFM of LSTM for athletes of different sports. Therefore, it is necessary to establish the estimation equation of LSTM or FFM of arms and legs for athletes. But in fact, for the research of BIA measurement equation of LSTM or FFM of athletes' arms or legs is very limited. Sardinha _et al._ [23] developed the estimation equation for limb LSTM for single-frequency BIA in Caucasian athletes. The results of the LSTM estimation equation in the VG of that study were that the LOA and correlation coefficient of the lower limb LSTM were −1.11 to 1.32 kg ($r = 0.90$) and −3.78 to 3.87 kg ($r = 0.94$), respectively. Using the resistance, reactance and anthropometric parameters of athletes in Taiwan as entered into the estimation equation of Sardinha _et al._

[23], the correlation coefficients between the estimated results of ArmsLSTM, LegsLSTM with DXA were $r = 0.81$ and $0.86$, respectively. The LOAs of Bland-Altman plots were $-2.70$ to $2.14$ kg and $-6.96$ to $2.32$ kg, respectively. The estimation equation established by Sardinha *et al.* [23] is better than the existing commercial BIA device in the measurement of LSTM or FFM of Arms and Legs. Therefore, it is necessary to establish an athlete-specific LSTM measurement equation for arms and legs.

This study and Sardinha *et al.* [23] both used stepwise regression analysis and the same candidate estimated variables. The ArmsLSTM measurement equation of this study was selected after stepwise regression analysis. The measured variables were $h^2/R$, Sex, Xc and h. The estimated variables of the LegsLSTM equation were $h^2/R$, h, Sex, Weight, Age and Xc. Compared with Sardinha *et al.*, the LegsLSTM equation in this study has more estimated variables for h, Age, and Xc. The BIA estimation equations ArmsLSTM$_{BIA-Asian}$ and LegsLSTM$_{BIA-Asian}$ obtained had LOA and correlation coefficients in the cross-validation group of $-1.53$ to $1.43$ kg ($r = 0.921$), $-2.68$ to $2.90$ kg ($r = 0.957$). Compared with the LSTM estimation equation developed by Sardinha *et al.*, the LSTM estimation equation developed in the present study is not only suitable for Asian athletes, but also has better overall performance in legs estimation. In this study, the estimated variables selected in sequence in the application of stepwise regression analysis were shown. It was shown in the results that the addition of Xc has little effect on the coefficient of determination of the estimated equation. But between Xc and LegsLSTM or ArmsLSTM respectively reached a moderate negative correlation, and will have a certain impact on reducing SEE.

Changes in body composition of the whole body or limb region may provide additional information on exercise science. In particular, it can be used to formulate an effective training program for different sports and positions. Therefore, in the future research direction, it is necessary to measure the body composition of each limb, especially the measurement of and research results for LSTM, in addition to the measurement of athletes' body composition. This prospective observational study applied supine bioimpedance measurements to validate the LSTM reported by Sardinha *et al.* [23] for extremities. Corrections for resistance and reactance were also performed before the experiment, and the study was replicated with reference to its experimental protocol.

Sex, sports, position, endurance sports, resistance sports or a mixture of the latter two will all affect the athlete's body composition. Because Sex has been added into the estimated variable in the stepwise regression analysis in the newly constructed estimation equation in this study. Therefore, this study has included both men and women and used the same equation to estimate their LSTM. At present, there are only a few studies on the body composition estimation equation of athletes' bioimpedance in a single sport [31]. In contrast, the scope of application of a single sport is also relatively limited. If the LSTM for athletes is added to the variable of their sport, or the sports are classified to construct and verify their LSTM estimation equations, it will definitely be more suitable for the athlete's composition measurement of their corresponding sports. Perhaps this could also be a good research direction in the future.

In the application of stepwise regression analysis, the estimation variables of the estimation equations entered first by ArmsLSTM or LegsLSTM were all $h^2/R$. Weight also achieves high correlation for ArmsLSTM or LegsLSTM. However, it can be observed from the correlation coefficient matrix, Weight and reactance are positively correlated with ArmLSTM, LegsLSTM and estimated variables, but Weight is still greater than reactance. This was also reflected in the selection order of LSTM estimation equations for estimated variables. In this study, if only the four parameters of age, height, weight, and sex were used as estimation variables, and ArmLSTM and LegLSTM were used as dependent variables, the $R^2$ and SEE of the estimation equation obtained through linear stepwise regression analysis were 0.790 and 1.020 kg, and

0.867 and 1.619 kg, respectively. When R and Xc were added to the estimation equation, the $R^2$ and SEE improved to 0.855 and 0.757 kg, and 0.916 and 1.431 kg, respectively. This demonstrates a significant improvement in the estimation accuracy of the equation. Additionally, $h^2$/R has a higher estimation or explanatory ability for ArmLSTM or LegLSTM compared to anthropometric predictions.

Regarding the selection of estimation variables in this study, it is important to consider not only the commonly used variables in bioimpedance analysis but also their physiological significance, the correlation between the selected estimation variables and their response variables, ease of measurement, measurement accuracy, reproducibility, and collinearity.

Athletes have a unique body composition [32] compared to non-athletes due to the requirements of competitive sports. Therefore, there may be larger errors when the LSTM measurement equation of BIA from non-athletes [33] are applied to athletes using. Relative compared with BIA used to measure the quantity of body composition, bioelectrical impedance vector analysis (BIVA) can be used to identify the subject's water and hydration status. Through the qualitative analysis of BIVA, it is possible to avoid make the relevant problem of BIA. Using BIVA can't estimate the body composition, but can draw the tolerance ellipses for specific ethnic groups such as athletes to evaluate the vector position. Qualitative analysis of BIVA can be used for comparing the physical characteristics [34, 35].

The equations established in this study are specific to the equipment and populations used. The use of BIA to measure body composition has a certain relationship with the geometric shape of the human body. Therefore, there may be certain estimation errors in the measurement of different special athletes. Furthermore, we used whole body rather than segmental impedance measurements. Therefore, in theory, the use of whole-body bioimpedance measurement devices to estimate athletes' LSTM may be inferior to that of segmental measurement devices. However, the results presented in this study are not worse than commercially available segmental bioimpedance measurement devices. It is also worth mentioning that it is a good choice to develop estimation equations for athletes of a single sport in the future. At the same time, it would be a better research direction if the differences of race can be considered. In addition, the study subjects were all elite athletes recruited in Taiwan whose data contributed to formulating the limbs LSTM estimation equations, which limits the extent to which study results can be generalized to other populations. Whether results of the present study are applicable to other races still needs to be verified in further studies. The goal of this study is to develop an LSTM estimation equation for Asian athletes using the bioimpedance analysis method. This will be compared with the LSTM estimation equation for European athletes using the same measurement method to elucidate the differences in the estimation equations between races. Validation and comparison of existing commercial bioimpedance devices against users' body composition or LSTM estimation results is a crucial area of research, particularly for athletes. This study could lead to further research in this direction.

A key objective of this study is to demonstrate that athletes of different races require race-specific samples to develop accurate estimation equations. The bioimpedance analysis method offers more valuable body composition measurements compared to anthropometric methods. Although the LSTM estimation equation published in this study cannot be used in clinical applications due to significant estimation errors, it still provides a convenient method for monitoring the LMST of athletes. This method is safe, non-invasive, and convenient.

In the future, based on this study, measuring the resistance and reactance of each limb at different frequencies and incorporating more anthropometric predictions should bring the method closer to clinical application.

## Conclusion

Single-frequency BIA estimation equations of LSTM for arms and legs of Asian athletes in Taiwan have been established and have demonstrated good performance, allowing them to quickly and accurately measure lean soft tissue mass of the arms and legs of Asian athletes. Nevertheless, BIA estimation equations of the limbs LSTM for Caucasian athletes should be carefully evaluated when applied to the measurement of the limb LSTM of Asian athletes. It is recommended that when Asian athletes measure the muscle mass of their limbs, they must use a specific and verified BIA estimation equation of limb LSTM, so that the measurement results can be actually used for reference value.

## Supporting information

**S1 File.**
(XLSX)

## Author Contributions

**Conceptualization:** Yeong-Kang Lai, Kuen-Chang Hsieh.

**Data curation:** Yeong-Kang Lai.

**Formal analysis:** Chu-Ying Ho, Hsueh-Kuan Lu.

**Funding acquisition:** Ai-Chun Huang.

**Investigation:** Chu-Ying Ho.

**Methodology:** Yeong-Kang Lai.

**Project administration:** Ai-Chun Huang, Kuen-Chang Hsieh.

**Supervision:** Kuen-Chang Hsieh.

**Validation:** Kuen-Chang Hsieh.

**Visualization:** Chu-Ying Ho, Hsueh-Kuan Lu.

**Writing – original draft:** Yeong-Kang Lai, Kuen-Chang Hsieh.

**Writing – review & editing:** Hsueh-Kuan Lu, Kuen-Chang Hsieh.

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
