## [Decision Letter · Decision Letter 0]

12 Apr 2024

PONE-D-24-08399Estimation equation of limb lean soft tissue mass in Asian athletes using bioelectrical impedance analysisPLOS ONE

Dear Dr. Hsieh,

Thank you for submitting your manuscript to PLOS ONE. After careful consideration, we feel that it has merit but does not fully meet PLOS ONE’s publication criteria as it currently stands. Therefore, we invite you to submit a revised version of the manuscript that addresses the points raised during the review process.

We look forward to receiving your revised manuscript.

Kind regards,

Gianpiero Greco

Academic Editor

PLOS ONE

Journal Requirements:

"This work was supported by grants from the Tzu Hui Institute of Technology Research Program (THIT-110004) and was supported in part by the National Science and Technology Council, Republic of China, under Grants MOST 111-2622-E-005-001."

"This work was supported by grants from the Tzu Hui Institute of Technology Research Program (THIT-110004) and was supported in part by the National Science and Technology Council, Republic of China, under Grants MOST 111-2622-E-005-001."

"This work was supported by grants from the Tzu Hui Institute of Technology Research Program (THIT-110004) and was supported in part by the National Science and Technology Council, Republic of China, under Grants MOST 111-2622-E-005-001."    

4. Thank you for stating the following in the Competing Interests/Financial Disclosure * (delete as necessary) section:

"The authors have read the journal’s policy and have the following competing interests: Dr. Kuen-Chang Hsieh was employed by a commercial company, StarBIA Meditek Co. Ltd., during this study. This does not alter our adherence to PLOS ONE policies on sharing data and materials. There are no patents, products in development or marketed products associated with this research to declare. All other authors have no conflicts of interest to declare. The funders had no role in the design of the study; in the collection, analyses, or interpretation of data; in the writing of the manuscript, or in the decision to publish the results."

We note that you received funding from a commercial source: "StarBIA Meditek Co. Ltd"

Additional Editor Comments :

Dear Authors,

please reply point by point to the reviewers' comments.

Many thanks!

Reviewers' comments:

Reviewer's Responses to Questions

**Comments to the Author**

1. Is the manuscript technically sound, and do the data support the conclusions?

Reviewer #1: Yes

Reviewer #2: Yes

2. Has the statistical analysis been performed appropriately and rigorously? 

Reviewer #1: No

Reviewer #2: Yes

3. Have the authors made all data underlying the findings in their manuscript fully available?

Reviewer #1: Yes

Reviewer #2: Yes

4. Is the manuscript presented in an intelligible fashion and written in standard English?

Reviewer #1: No

Reviewer #2: Yes

5. Review Comments to the Author

Reviewer #1: Dear Authors

the manuscript describe the validation of a new equation to predict limb lean soft tissue mass from BIA assessment in asian athletes.

This is my comment:

- in general the introduction appear too dispersive and the focus on the aim of the study is only in the end of the section. Much more references are used, some appear dated and others not relevant to the sentence (i.e. ref. 5).

- pag 2 line 66: please check "of muscle and muscle volume (MV),"

- is necessary a reference at pag. 4, line 102 "There are certain differences in the physiological quality of different races. Therefore, it is necessary to establish and verify the measurement equations of LSTM and BIA for Asian athletes."

- There is the use of "racial" terminology. Probably "ethnicity" is more appropriate. In addition, there are no information about the different socio-cultural habits in different populations instead the ethnicity-anthropological approach used in this manuscript.

- Why the authors choose to perform the assessments at 1:00 p.m? Before or after the lunch?

- no reference were reported for reliability of the instrumentation used in the manuscript.

- table 2 and table 3 is not clear. Please ensure the presentation.

- The authors performed a comparison of Sardinha equation. However, the Sardinha study was performed with a different BIA device. Please stated this aspect in discussion section and justify this choice.

- fig. 2 and 3 not reported any statistical results. Please add some additional informations.

- Why the authore not perfomed the ROC analysis as in 10.1186/s12967-023-04795-z

- please add some comment about the results described in fig. 3 b. and 3 d. Please clarify.

Reviewer #2: Abstract and elsewhere in the manuscript. The authors must make it clear that generated prediction equations are only applicable to measurements obtained at the locations used by the InBody device. These are unusual and not the commonly used wrist-ankle locations or the sole and palm as used in conventional stand-on devices.

Line 70 Strictly, BIA is considered a two-compartment model; a conductive compartment (FFM)and a non-conductive compartment (fat). It is a predictive or estimating method. It does NOT measure BMC although BIA manufacturers frequently provide a doubly indirect estimate. Please correct to describe BIA accurately as a 2C system.

Line 71 "non-penetrating" I believe that you mean non-invasive.

Line 86 onward. Again please avoid suggesting that BIA MEASURES body composition, e.g. line 9. It PREDICTS or ESTIMATES BC.

Line 146 Please make it clear whether you are simply making whole body impedance measurements or that the device measures segments.

Line 148. Why was this analyser used when the authors had available an alternative analyser (line 137). Why was the opportunity not taken to compare predictions from the two commercial analysers. In addition to testing their own equation? This would have added to the study.

Line 171. Were the ROI those provided by the software or adjusted/optimised by the DXA operator?

Lines 210-216. Please highlight some key observations from these data, e.g., a male female difference, apparent proportional effect in the LOA plots. At present, essentially fig titles are simply presented.

Line 217. Please state what was the basis for variable selection. Those chosen are typical for this type of study but their inclusion should be justified.

Line 218. Sex not gender.

Line 240 What is this reproducibility measure? How was it calculated? Does it have units?

Line 251. This picks up my first point. That empirically-derived prediction equations are population/device specific. Also highlights my point about why you did not test your Tanita predictions.

Line 284.and elsewhere. You quote absolute LOA, e.g., approximately +/-2.8 kg for leg LSTM. Since leg LSTM is around 19 kg, these LOA equate to about +/-15%. In practice the question is whether this level of inaccuracy is clinically acceptable? Would an athlete or coaches be happy with an estimate of LSTM with +/-15% LOA error. This point must be addressed. Your equation may perform better than the Sardinha equation but it may still not be sufficiently accurate for practical use.

Line 312. It is often found that inclusion of BIA in algorithms only improves prediction marginally compared to simple anthropometric predictions. Was this actually tested? If not why not?

Line 327 Include in limitations that the equation is specific to this device and population.

Line 347

What do you mean here by "proprietary"? Are you suggesting manufacturer's predictions?

6. PLOS authors have the option to publish the peer review history of their article (what does this mean?). If published, this will include your full peer review and any attached files.

Reviewer #1: No

Reviewer #2: No

---

## [Author Response · Author response to Decision Letter 0]

1 Jun 2024

Reviewer #1: Dear Authors

the manuscript describe the validation of a new equation to predict limb lean soft tissue mass from BIA assessment in asian athletes.

This is my comment:

- in general the introduction appear too dispersive and the focus on the aim of the study is only in the end of the section. Much more references are used, some appear dated and others not relevant to the sentence (i.e. ref. 5).

Response: 

Thank you for your suggestion. We have revised this and deleted the narrative on lines 58-63 in the manuscript to make it more narrative, rigorous and easier to read, as shown in the revised manuscript. The above modifications streamline redundant words and also de-cite outdated documents.

- pag 2 line 66: please check "of muscle and muscle volume (MV),"

Response:

Thank you for the reminder, we have revised this statement to "Although CT and MRI provide information on muscle cross-sectional areas (CSA) and muscle volume (MV)," as shown in the revised manuscript.

- is necessary a reference at pag. 4, line 102 "There are certain differences in the physiological quality of different races. Therefore, it is necessary to establish and verify the measurement equations of LSTM and BIA for Asian athletes."

Response:

Thank you for your suggestion, we have added the appropriate reference [30], and revised its description to " More cross-ethnic population studies on body composition are warranted. These studies should be meticulously conducted, employing suitable methodologies and standardization, to ensure the value of the gathered information [30].

30. Deurenberg P, Deurenberg-Yap M. Validity of body composition methods across ethnic population groups. Acta Diabetologica. 2003; 40;S246-S249.

- There is the use of "racial" terminology. Probably "ethnicity" is more appropriate. In addition, there are no information about the different socio-cultural habits in different populations instead the ethnicity-anthropological approach used in this manuscript.

Response:

Thank you for your suggestion, we have revised "racial" to "ethnicity".

- Why the authors choose to perform the assessments at 1:00 p.m? Before or after the lunch?

Response:

Thank you for your reminder, 1:00 pm is the benchmark point for a unified experiment, “and there is no lunch to maintain a fasting state before measurement." This description has been added to the Methods section, line 126-127.

- no reference were reported for reliability of the instrumentation used in the manuscript.

Response:

Thank you for your suggestions and reminders, " The InBody S-10 is a Class II medical device approved by the FDA (Regulation Number: 870.2770).", this description has been added to the Methods section, line 144-145.

- table 2 and table 3 is not clear. Please ensure the presentation.

Response:

Thank you for your reminder and suggestions. " The estimated variables selected in the stepwise regression analysis method, along with the corresponding regression coefficients, intercept, SEE, determination coefficient, variance inflation factor, and standardized coefficient of the estimation equation, are shown in Tables 2 and 3. In the stepwise regression analysis method, ArmLSTM was used as the response variable. : "," Table 2 and Table 3 have shown the choosing estimation variables and their orders when evaluating ArmsLSTM and LegsLSTM." This description has been added to the Results section, line 2223-226.., line 238-239 respectively.

- The authors performed a comparison of Sardinha equation. However, the Sardinha study was performed with a different BIA device. Please stated this aspect in discussion section and justify this choice.

Response:

Thank you for your suggestions and reminders. " The resistance and reactance measurement environments required for the Sardinha equation compared in this study were also supine with adhesive electrodes. Although different from the BIA device (BIA 101 Anniversary AKERN/RJL Systems; Florence, Italy) used by Sardinha et al. [23], the InBody S-10 used in this study underwent standard resistance and reactance correction procedures. This ensures the accuracy of the measured resistance and reactance.” This narrative has been added to the revised manuscript in the discussion section, line 288-293.

- fig. 2 and 3 not reported any statistical results. Please add some additional informations.

Response:

Thank you for your reminder and suggestions. ” Figures 2a and 2c are scatter plots of ArmsLSTM and LegsLSTM estimates compared to DXA results using the Sardinha equation. The regression line (thin solid line) shows a weak fit with the equivalence line (dashed line), indicating that the accuracy of the Sardinha equation for estimating ArmsLSTM and LegsLSTM in Asian athletes is limited. The Bland-Altman plots in Figures 2b and 2d show the limits of agreement (LOA) for ArmsLSTM and LegsLSTM estimates using the Sardinha equation in Asian athletes. The trend line (dashed line) in the plots indicates a significant proportional bias between the two measurement methods.

Figures 3a and 3c show scatter plots of ArmsLSTM and LegsLSTM estimates compared to DXA results for the validation group. The regression line (thin solid line) closely aligns with the equivalence line (dashed line). The Bland-Altman plots in Figures 3b and 3d represent the ArmsLSTM and LegsLSTM estimation equations established by the model group. This indicates no proportional error in the validation group's estimated results. The limits of agreement (LOA) in Figures 3b and 3d are significantly narrower than those in Figures 2b and 2d." This description is added to the Result section, line 216-222, line 244-249.

- Why the authore not perfomed the ROC analysis as in 10.1186/s12967-023-04795-z

Response:

Thank you for your suggestions and reminders. The main purpose of this study is to confirm the applicability of the established LMST for athletes of different races, and also to establish equations suitable for Asian athletes for estimating LSTM of upper and lower limbs. In addition, the models and verification groups are used for verification. Use statistical techniques such as stepwise regression analysis, Paired-t test, Lin’s concordance correlation coefficient, Bland-Altman Plots, intraclass correlation coefficient and other statistical techniques to conduct data analysis. These indicators and methods are the same as the statistical analysis in Sardinha equation [23], which is more conducive to comparison between this study and Sardinha equation [23]. The above indicators have included most of the statistical analysis tools and result interpretation required for this study. 

- please add some comment about the results described in fig. 3 b. and 3 d. Please clarify.

Response: 

Thank you for your reminder. As suggested above, we have added the supplementary description to the Results Section, Line 244-249.

 

Reviewer #2: 

Abstract and elsewhere in the manuscript. The authors must make it clear that generated prediction equations are only applicable to measurements obtained at the locations used by the InBody device. These are unusual and not the commonly used wrist-ankle locations or the sole and palm as used in conventional stand-on devices.

Response:

Thank you for your suggestions and reminders. We have strengthened the description of the measurement methods and limitations in the Abstract, Methods, and Discussion of the manuscript. The description " Using the InBody S-10, resistance and reactance were measured at 50 kHz from the right palm to the right sole while the participant was in the supine position. " was added to the Abstract. " Resistance and reactance were measured from the dorsum of the right palm to the dorsum of the right foot using a bioimpedance analyzer, the S-10 (InBody Co., Ltd., Seoul, South Korea), operating at a frequency of 50 kHz while the participant was in a supine position. " was added to the Methods. " Most existing commercial multi-limb bioimpedance analyzers are designed for use in a standing position and are equipped with stainless steel electrode plates to measure the resistance and reactance of the limbs and trunk at different frequencies. In this study, the resistance and reactance of a single section from the dorsum of the right palm to the dorsum of the right foot were measured in a supine position using electrodes." was added to the Discussion section.

Line 70 Strictly, BIA is considered a two-compartment model; a conductive compartment (FFM)and a non-conductive compartment (fat). It is a predictive or estimating method. It does NOT measure BMC although BIA manufacturers frequently provide a doubly indirect estimate. Please correct to describe BIA accurately as a 2C system.

Response: 

Thank you for your reminder and suggestions. As stated here, the three-compartment model has been revised to the two-compartment model.

Line 71 "non-penetrating" I believe that you mean non-invasive.

Response:

Thank you for your suggestions and reminders, we have revised it to non-invasive.

Line 86 onward. Again please avoid suggesting that BIA MEASURES body composition, e.g. line 9. It PREDICTS or ESTIMATES BC.

Response:

Thank you for your suggestions and reminders. In addition to revising this to “estimate”, we have also revised other similar issues in the manuscript.

Line 146 Please make it clear whether you are simply making whole body impedance measurements or that the device measures segments.

Response:

Thank you for your reminders and suggestions. We have revised or supplemented the Introduction section and Abstract section respectively to make readers more intuitively understand that this study was to establish the impedance measurement mode of the whole body.

Line 148. Why was this analyser used when the authors had available an alternative analyser (line 137). Why was the opportunity not taken to compare predictions from the two commercial analysers. In addition to testing their own equation? This would have added to the study.

Response:

Thank you for your suggestions and reminders. " The goal of this study is to develop an LSTM estimation equation for Asian athletes using the bioimpedance analysis method. This will be compared with the LSTM estimation equation for European athletes using the same measurement method to elucidate the differences in the estimation equations between races. Validation and comparison of existing commercial bioimpedance devices against users' body composition or LSTM estimation results is a crucial area of research, particularly for athletes. This study could lead to further research in this direction. " The above description has been added to the Discussion section, line 384-390.

Line 171. Were the ROI those provided by the software or adjusted/optimised by the DXA operator?

Response:

Thank you for your suggestions and reminders. “The regions of interest (ROI) for the whole body, trunk, legs, arms, android, and gynoid were automatically determined by the software.” This statement has been added to the Methods section, Line 169-170 in the revised manuscript.

Lines 210-216. Please highlight some key observations from these data, e.g., a male female difference, apparent proportional effect in the LOA plots. At present, essentially fig titles are simply presented.

Response:

Thank you for your suggestion." This study used data measured from Asian athletes and applied the equation of Sardinha et al. For the upper limbs, the LSTM of female athletes was generally overestimated, while the LSTM of male athletes was slightly underestimated. For the lower limbs, the LSTM of male athletes was significantly underestimated. These results showed clear proportional effects, with greater underestimation as the LSTM of the upper and lower limbs increased. Therefore, the applicability of the LSTM estimation equation established for European athletes to Asian athletes must be carefully evaluated." This description has been added to the Discussion section, line 267-273.

Line 217. Please state what was the basis for variable selection. Those chosen are typical for this type of study but their inclusion should be justified.

Response:

Thank you for your reminder and suggestions. " Regarding the selection of estimation variables in this study, it is important to consider not only the commonly used variables in bioimpedance analysis but also their physiological significance, the correlation between the selected estimation variables and their response variables, ease of measurement, measurement accuracy, reproducibility, and collinearity.” This description was added to the Discussion section, line 358-361 in the revised manuscript.

Line 218. Sex not gender.

Response:

Thank you for your reminder and suggestions. In addition to correcting this, we have also corrected other similar problems in the manuscript, as shown in the revised manuscript.

Line 240 What is this reproducibility measure? How was it calculated? Does it have units?

Response:

Thank you for your suggestions and reminders. " Intraclass correlation coefficient (ICC) analysis was conducted using two repeated measurements to assess the reproducibility of estimating ArmsLSTM and LegsLSTM in this study. The measurement unit was kilograms (kg)." This description has been added to the revised manuscript , Statistical analysis section, Line 196-198.

Line 251. This picks up my first point. That empirically-derived prediction equations are population/device specific. Also highlights my point about why you did not test your Tanita predictions.

Response:

Thank you again for your suggestions and reminders. Regarding your suggestion "to explore the verification of LSTM measured by commercial body composition analyzers in athletes", I also think it is a very important research direction. The Tanita BC-418MA used to measure body weight in this study was discontinued approximately ten years ago. The Inbody S-10 used in this study to measure resistance and reactance is believed to have more research value if it is used to verify the estimation results of athletes' body composition against commercial BIA devices such as the commonly used and standing Inbody 770. This is also our future research direction.

Line 284.and elsewhere. You quote absolute LOA, e.g., approximately +/-2.8 kg for leg LSTM. Since leg LSTM is around 19 kg, these LOA equate to about +/-15%. In practice the question is whether this level of inaccuracy is clinically acceptable? Would an athlete or coaches be happy with an estimate of LSTM with +/-15% LOA error. This point must be addressed. Your equation may perform better than the Sardinha equation but it may still not be sufficiently accurate for practical use.

Response:

Thank you for your reminder and suggestions. " A key objective of this study is to demonstrate that athletes of different races require race-specific samples to develop accurate estimation equations. The bioimpedance analysis method offers more valuable body composition measurements compared to anthropometric methods. Although the LSTM estimation equation published in this study cannot be used in clinical applications due to significant estimation errors, it still provides a convenient method for monitoring the LMST of athletes. This method is safe, non-invasive, and convenient. 

In the future, based on this study, measuring the resistance and reactance of each limb at different frequencies and incorporating more anthropometric predictions should bring the method closer to clinical application." The above narrative has been added to Discussion section, Line 391-399, as shown in revised manuscript.

Line 312. It is often found that inclusion of BIA in algorithms only improves prediction marginally compared to simple anthropometric predictions. Was this actually tested? If not why not?

Response:

Thank you for your suggestion. " In this study, if only the four parameters of age, height, weight, and sex were used as estimation variables, and ArmsLSTM and LegsLSTM were used as dependent variables, the R² and SEE of the estimation equation obtained through linear stepwise regression analysis were 0.790 and 1.020 kg, and 0.867 and 1.619 kg, respectively. When R and Xc were added to the estimation equation, the R² and SEE improved to 0.855 and 0.757 kg, and

---

## [Decision Letter · Decision Letter 1]

26 Jun 2024

Estimation equation of limb lean soft tissue mass in Asian athletes using bioelectrical impedance analysis

PONE-D-24-08399R1

Dear Dr. Hsieh,

We’re pleased to inform you that your manuscript has been judged scientifically suitable for publication and will be formally accepted for publication once it meets all outstanding technical requirements.

Kind regards,

Gianpiero Greco

Academic Editor

PLOS ONE

Additional Editor Comments (optional):

Reviewers' comments:

Reviewer's Responses to Questions

**Comments to the Author**

1. If the authors have adequately addressed your comments raised in a previous round of review and you feel that this manuscript is now acceptable for publication, you may indicate that here to bypass the “Comments to the Author” section, enter your conflict of interest statement in the “Confidential to Editor” section, and submit your "Accept" recommendation.

Reviewer #2: All comments have been addressed

2. Is the manuscript technically sound, and do the data support the conclusions?

Reviewer #2: Yes

3. Has the statistical analysis been performed appropriately and rigorously? 

Reviewer #2: Yes

4. Have the authors made all data underlying the findings in their manuscript fully available?

Reviewer #2: Yes

5. Is the manuscript presented in an intelligible fashion and written in standard English?

Reviewer #2: Yes

6. Review Comments to the Author

Reviewer #2: The authors have satisfactorily addressed the issues raised in my original review.

7. PLOS authors have the option to publish the peer review history of their article (what does this mean?). If published, this will include your full peer review and any attached files.

Reviewer #2: No

---

## [Editor Report · Acceptance letter]

17 Jul 2024

PONE-D-24-08399R1 

PLOS ONE

Dear Dr. Hsieh, 

I'm pleased to inform you that your manuscript has been deemed suitable for publication in PLOS ONE. Congratulations! Your manuscript is now being handed over to our production team.

Kind regards, 

on behalf of

Dr. Gianpiero Greco 

Academic Editor

PLOS ONE